# Evolutionary Exploration with LLM-Generated Influenza Variants

Rob Dunne[1], Suk Yee Yong[1], Carol Lee[1], Alex McAuley[1]

Commonwealth Scientific and Industrial Research Organisation (CSIRO), Australia.

## 1   Introduction and Motivation

Current influenza surveillance only identifies dangerous viral variants after they emerge and spread. The laboratory assays are labor-intensive, requiring weeks to months, and do not work reliably with all variants [1]. However viral genome sequencing is now significantly cheaper and faster than serological testing; however, predicting which sequences represent pandemic threats remains challenging.

Foundation models present a unique opportunity to generate synthetic viral sequences representing evolutionary trajectories. Unlike typical language model applications seeking single "correct" answers, viral evolution requires understanding entire distributions of variants. Real viral populations are heterogeneous, surveillance systems must identify concerning subsets within populations, and single sequence predictions miss evolutionary breadth.

**Our Approach:** We fine-tune genSLM (25 billion parameters, trained on 110 million prokaryotic sequences [2]) using codon-level tokenization on influenza hemagglutinin sequences. The constrained vocabulary (69 vs 50,000+ tokens in text models) and built-in biological constraints (reading frames, codon usage patterns) enable more interpretable generation with experimental validation capability. We extend the analysis from analyzing single sequences to ranking thousands of variants by pandemic risk.

## 2   Dual Generation Strategies

### 2.1   Temperature-Based Generation: The GPT Analogue

Our first approach employs autoregressive generation paralleling GPT's text generation, predicting the next codon given previous codons using standard sampling techniques:

- **Temperature scaling**: Controls randomness (0.1-0.5 conservative, 1.0-2.0 diverse)
- **Top-k/top-p filtering**: Limits consideration to most probable codons
- **Target mutation rate**: Guides expected changes from reference

This method suits routine surveillance, modeling seasonal drift patterns of 1-8 mutations per year while maintaining sequence length and biological plausibility.

## 2.2   Reward-Guided Beam Search: RLHF for Biology

Our second approach implements a reward-guided beam search, analogous to Reinforcement Learning from Human Feedback (RLHF) techniques used to align GPT. Instead of optimizing for helpfulness in text, we optimize viral sequences for biological properties:

$$\text{Score} = \mu \times P_{\text{model}} + (1 - \mu) \times R(x) \tag{1}$$

where $P_{\text{model}}(x|\theta) = \prod_{i=1}^{L} P(x_i|x_{1:i-1}, \theta)$ for sequence length $L$, $\mu$ balances model probability and reward function $R(x)$, analogous to Kullback–Leibler ) divergence penalties preventing nonsense while pursuing rewards. Multi-objective rewards include:
  - **alignment**: Similarity to reference (prevents biological irrelevance)
  - **plausibility**: Realistic mutation rates using global alignment scores
  - **diversity**: Population-level sequence variety

This enables insertions/deletions alongside point mutations, modeling pandemic emergence scenarios with larger evolutionary leaps.

# 3   Population-Centric Evaluation Framework

## 3.1   The Assessment Challenge

Unlike standard LLM applications seeking single "best" answers, viral sequence generation requires understanding distributions. We transform the question from "What is the next variant?" to "What does the landscape of possible variants look like, and which regions pose pandemic risk?"

## 3.2   NIAViD Physicochemical Scoring

We apply NIAViD (Novelty in Antigenic Variant Detection) using five physicochemical properties mechanistically relevant to viral antigenicity [1]:
  - Boman's Index (protein binding potential),
  - isoelectric point (charge neutrality pH),
  - hydrophobicity (water repulsion affecting epitope exposure),
  - electrostatic charge (immune complex formation), and
  - instability index (protein stability).

These properties are calculated for HA1 sequences and then subjected to anomaly detection, using a 5 year sliding window as the training data.

## 3.3   Biological Plausibility with BLOSUM62

We integrate BLOSUM62 substitution matrices assessing evolutionary acceptability of amino acid changes. Conservative changes (BLOSUM $\geq 0$) like K→R are likely tolerated, moderate changes (-1 to -2) like K→E show possible functional impact, while radical changes ($<$-3) like G→W indicate likely structural disruption.

Combined quality assessment integrates BLOSUM with dN/dS ratios, epitope targeting, and synonymous mutation rates:

$$\text{Quality} = 0.3 \times \mathbb{I}(\text{dN/dS} \in [0.5, 3.0]) + 0.3 \times \frac{\text{avg\_BLOSUM} + 4}{8} +$$
$$0.2 \times \mathbb{I}(\text{epitope}) + 0.2 \times \mathbb{I}(\text{syn\_rate} > 0.2)$$

Scores above 0.7 indicate high biological plausibility.

## 3.4 Population-Level Risk Stratification

Our pipeline generates 100-1000 synthetic variants, scores populations with NIAViD and BLOSUM62, and identifies high-risk subsets (top 10% novelty scores) for further consideration. Population metrics include novelty distribution, extreme outlier identification, and mutation quality patterns.

# 4 Results

## 4.1 Generation Method Validation

The gensLM model was finetrained with 40000 sequences (training), and each for the test and validation sets. Temperature-based generation produces realistic characteristics: 0.1-0.5% mutation rates matching annual drift, 30-50% synonymous mutations (biologically realistic), average BLOSUM scores -0.5 to +1.0 (conservative to moderate), and 100% reading frame maintenance. Reward-guided beam search enables controlled exploration with adjustable mutation rates (0.05-0.3%), insertion/deletion capability, and property targeting across broader sequence space.

## 4.2 NIAViD Population Scoring

We first generate variant populations, then score each sequence with NIAViD for novelty and BLOSUM62 for biological plausibility, creating integrated quality scores. Key findings include: baseline populations from recent seasonal strains show 5-15% variants as anomalous; pandemic-like populations from pre-2009 sequences show 40-60% variants with characteristics similar to actual 2009 emergence; 1000 variants analyzed in under 5 minutes; 85% consensus agreement between algorithms. Historical validation confirms biological relevance, see figure 1 where the 2009 H1N1 flu season is clearly recoved. The data for 2021 are synthetic mutations based on a single sample from 2020 and evalauded with the NIAViD 5 year sliding window.

## 4.3 Practical Applications

This framework enables: (1) proactive surveillance identifying concerning variants before natural emergence, (2) vaccine strain selection based on predicted antigenicity, (3) focused laboratory validation on highest-risk variants, and (4)

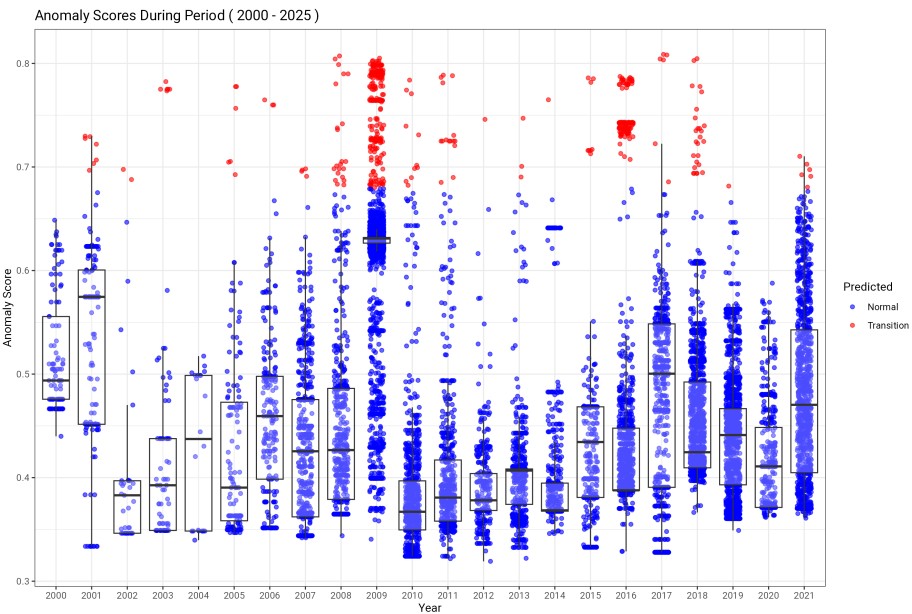

**Fig. 1.** NIAViD 5 year sliding window showing synthetic sequence scoring (generated using temperature-based method). The method pick up the H1N1 flu season in 2009. The data for 2021 are synthetic mutations based on a single sample from 2020.

pandemic preparedness through population-based analysis. The approach transforms assessment from single sequence prediction to population-based risk stratification.

# 5    Conclusion

We present the first application of reward-guided generation to viral surveillance, establishing a population-centric evaluation framework for biological sequences. This work demonstrates applying RLHF-style techniques to biological domains with objective validation criteria, transforming variant assessment from single predictions to population-based risk stratification. Our framework bridges modern AI capabilities with practical public health needs, establishing new paradigms for AI-augmented pandemic preparedness where distributions matter more than individual predictions.

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
