# OpenReview forum: "Evolutionary Exploration with LLM-Generated Influenza Variants"
_AJCAI/2025/Workshop/AIML-CEB — AIML-CEB 2025 Oral_

### Official Review · Reviewer_Ay7w · 2025-11-09
**Optimizing LLM-Based Generation of Influenza Variants**

**Rating:** 9
**Confidence:** 4

**Review:**

This paper fine-tunes GenSLM (25B) on influenza hemagglutinin sequences with codon-level tokenization, then optimizes generation via two modes: (i) a temperature-based strategy (temperature, top-k/top-p, and a target mutation rate to mirror seasonal drift), and (ii) a reward-guided beam search that trades off model likelihood and multi-objective biological rewards (alignment, realistic mutation rates, and population-level diversity). The authors further propose a population-centric evaluation that integrates NIAViD physicochemical anomaly detection with BLOSUM/dN/dS/epitope constraints to better reflect biological plausibility and risk at the population level.

Pros:

•	Optimizes sequence generation to better align with biological plausibility and seasonal drift patterns.

•	Introduces reward-guided generation (RLHF-style) to biological sequence design/surveillance with clear, interpretable objectives.

Cons:

•	Missing ablations against (a) an unfine-tuned GenSLM and (b) naïve sampling without the proposed generation strategies; including these would clarify where gains come from (pretraining vs. fine-tuning vs. decoding).

---

### Official Review · Reviewer_pFva · 2025-11-09
**LLM-generated influenza variants for evolutionary exploration**

**Rating:** 7
**Confidence:** 3

**Review:**

The authors seek to address the limitations of haemagglutination inhibition assays with an LLM-based sequence generation and stratification strategy for predicting the landscape of influenza variants and their pandemic risk. They introduce a novel reward-guided beam search strategy for generating viral sequences and present a framework for evaluating predicted variants. The paper is well-written, with the sequence generation and scoring strategies well described. They also show that their strategy was able to predict the H1N1 influenza season in 2009.

Understanding that there was a page limit for submissions to this venue, the following could be clarified to strengthen the work for submission to future venues:

* What is the size of the influenza finetuning dataset and what finetuning strategy was used?
* The authors state that the recent H3N2 viruses do not always agglutinate RBCs, which perhaps suggests that influenza viruses do not all contain haemagglutinin, particularly those recently evolved. It is confusing that they then finetune genSLM on haemagglutinin sequences with the objective of predicting future influenza variants. Is there an alternative protein that is constitutively expressed in influenza variants, or can entire viral genomes be used for finetuning instead?
* Are the synthetic variants shown in Fig. 1 generated using the temperature-based or beam search-based generation strategy?
* The high-risk generated variants are identified by taking the top 10% novelty scores. How are the novelty scores calculated? Presumably, highly novel sequences may not be biologically plausible, so are the candidate sequences first filtered for biological plausibility and physicochemical properties prior to ranking novelty?
* Was there benchmarking to compare temperature-based and beam search-based sequence generation strategies?
* Clarify the link between the advances in this work to the claim of improving pandemic preparedness.

---

### Official Review · Reviewer_SELQ · 2025-11-11
**Vague language makes it difficult for the AI AND biology non-specialists.**

**Rating:** 7
**Confidence:** 3

**Review:**

This is a very jargon-heavy Abstract that makes it difficult to understand the links between concepts that logically give rise to a model that has potential for real-world impact. I think it is "We apply genomic measures transform variant assessment from single sequence prediction to population-based risk stratification" but I cannot tell exactly what this means in a practical sense. While I like the balance between the AI focus and the biological validation (the latter is also vague), the path from framework to real-world impact is obscured by a great deal of assumed expertise in both spaces, making this somewhat inaccessible.

From a data perspective, it would be good to have seen up front how many influenza hemagglutinin sequences were used for the fine tuning, and whether there was any exploration of the effect of reducing the fine-tuning data to see whether there was any measurable reduction in the model's effectiveness. No reduction might suggest it is sufficient data for other similar generative exercises in other viruses with pandemic potential - a useful finding on its own.

I would suggest to the authors they should take care to explain in plain terms what the particularly jargon-y concepts are (ie hemagglutination inhibition assays, actionable surveillance intelligence, temperature-based autoregressive generation, reward-guided beam search enabling insertions/deletions) so the reader can draw a mental map with as little abstraction as possible to help them understand the model and its potential for impact.

---

### Decision · Program_Chairs · 2025-11-12

Accept (Oral)